# Efficacy of a Modified Clinoptilolite Based Adsorbent in Reducing Detrimental Effects of Ochratoxin A in Laying Hens

**DOI:** 10.3390/toxins13070469

**Published:** 2021-07-07

**Authors:** Marko Vasiljević, Darko Marinković, Dragan Milićević, Jelka Pleadin, Srđan Stefanović, Saša Trialović, Jog Raj, Branko Petrujkić, Jelena Nedejković Trialović

**Affiliations:** 1The Faculty of Veterinary Medicine, University of Belgrade, Bulevar oslobodjenja 18, 11000 Belgrade, Serbia; marko.vasiljevic@patent-co.com; 2Department of Pathology, Faculty of Veterinary Medicine, University of Belgrade, Bulevar oslobodjenja 18, 11000 Belgrade, Serbia; darko@vet.bg.ac.rs; 3Institute of Meat Hygiene and Technology, Kaćanskog 13, 11040 Belgrade, Serbia; dragan.milicevic@inmes.rs (D.M.); srdjan.stefanovic@inmes.rs (S.S.); 4Croatian Veterinary Institute, Laboratory for Analytical Chemistry, Savska cesta 143, 10000 Zagreb, Croatia; pleadin@veinst.hr; 5Department of Pharmacology and Toxicology, Faculty of Veterinary Medicine, University of Belgrade, Bulevar oslobodjenja 18, 11000 Belgrade, Serbia; sasa@vet.bg.ac.rs; 6Patent Co, DOO., Vlade Ćetkovića 1A, 24211 Mišićevo, Serbia; jog.raj@patent-co.com; 7Department of Animal Nutrition and Botany, Faculty of Veterinary Medicine, University of Belgrade, Bulevar oslobodjenja 18, 11000 Belgrade, Serbia; petrujkic@vet.bg.ac.rs

**Keywords:** adsorbent, OTA, performance traits, pathohistological changes, OTA residue, eggs, risk assessment

## Abstract

Background: The objective of this study was to evaluate the efficacy of modified clinoptilolite (Minazel Plus^®^, MZ) as a mycotoxin adsorbent for preventing the negative the effects of ochratoxin A (OTA) on performance, pathohistological changes, and OTA residue in the eggs of laying hens. Methods: Forty eight (*n* = 48) laying hens (27 weeks old) were equally divided into six groups and depending on the type of addition were allocated to the following experimental treatments for 7 weeks: E-I group-1 mg/kg OTA; E-II group 0.25 mg/kg OTA; E-III group 1 mg/kg OTA + 0.2% of MZ; E-IV group 0.25 mg/kg OTA + 0.2% of MZ; MZ group supplemented with 0.2% of the adsorbent; and control (K, without feed additive). Results: Overall, the addition of 0.2% MZ to laying hen feed mitigated the harmful effects of OTA on target organs and reduced the presence of OTA residue in eggs. The groups that received 0.2% of MZ achieved better production results in terms of body weight, number of eggs, and feed consumption, compared to the other treatments. Conclusions: The current findings confirm the efficacy of MZ in preventing performance losses in laying hens exposed to OTA, as well as for improving the welfare and health of food producing animals.

## 1. Introduction

Mycotoxins are products of the secondary metabolism of certain filamentous fungi. These fungal species, when present in food and feed in sufficiently high levels, may cause serious diseases in animals and humans. Of the thousands of mycotoxins, OTA is one of the best known, and with the greatest public health and agro-economic significance due to its worldwide occurrence and prevalence [1]. OTA is primarily produced during storage by several fungal strains of *Aspergillus* and *Penicillium species* under varying environmental conditions (in tropical, warmer, or colder regions) [2]. OTA is considered a potent nephrotoxic mycotoxin that causes renal toxicity and possesses carcinogenic, teratogenic, immunotoxic, and possibly neurotoxic properties, based on experimental animal studies [1]. Thus, OTA has been classified by the IARC as a possible human carcinogen (Group 2B) [3].

In animals, mycotoxins produce a broad range of harmful effects on livestock health, production, and welfare, resulting in significant economic losses [4]. Mycotoxins cause economic losses, either directly due to animal’s death, or indirectly due to reduction in animal productivity, increased incidence of disease due to immuno-suppression, damage to vital organs accompanied by pathological change, interference with reproductive performance, and reduced or no response to veterinary therapies [5]. In poultry, after a prolonged OTA intake at a level of 0.5 mg OTA/kg in feed, signs of chronic ochratoxicosis linked to renal diseases occur, followed by a decrease in feed intake and disorders in egg production and egg quality [6]. It is important to emphasize that in field conditions non-specific signs of mycotoxicosis associated with the presence of multiple contributing factors occur, making it difficult to establish a clinical diagnosis. OTA has the potential to bioaccumulate in the organism, and thus consumption of OTA by birds may also be associated with the appearance of OTA residues in edible tissues and eggs [7,8], which might affect the safety of egg production throughout the food chain. Among food commodities, little is known about the risks posed by exposure to mycotoxins by consuming poultry products in Serbia [9].

In temperate countries such as Serbia, OTA is produced by *P. verrucosum,* which has been found to be associated with contamination of several foods and feedstuffs. Thus, OTA is one of the most studied mycotoxins in this region. Recently studies have confirmed that prolonged drought favored the growth of certain *Aspergillus* species and synthesis of OTA [10]. Extreme weather events in Serbia pose one of the greatest risks for the contamination of cereals such as wheat, maize, barley, and oat by various species of ochratoxigenic fungi and their related metabolites [11]. Milicevic et al. [12] suggested a set of prevention and control measures to mitigate the negative effects of mycotoxins on animal health, quality, and the safety of their products. In intensive poultry production various mineral adsorbents are often used to mitigate the harmful effects of mycotoxins [13,14]. Minazel Plus^®^ (MZ) is an adsorbent created as a result of an ion exchange reaction between inorganic cations on the mineral surface and organic cations. The addition of organic cations is used to change the mineral surface. The result of this addition is not a simple mixture of mineral and organic phases, but a completely new compound that is an organic complex. New active centers, which are formed on the mineral surface, ensure efficient binding, not only of polar mycotoxins (Aflatoxins, Ergot Alkaloids, etc.), but also of non-polar mycotoxins (Zearalenone, Ochratoxin A, T-2 toxin, etc.). 

The aim of this study was to assess the effect of a locally available, low-cost, and patented modified clinoptilolite adsorbent (MZ) for reducing the adverse effects on growth performance and target tissues, as well as on egg quality parameters and OTA residues in the eggs of laying hens fed with OTA-contaminated feed. In addition to determine the protective effects of the adsorbent, the presence of OTA residues in eggs during the in vivo experimental trial was evaluated, with a hypothetical estimation of human daily intake (EDI) of OTA by consuming eggs contaminated with OTA. The results of our study are expected to raise awareness of the health risks associated with the presence of OTA in the feed of laying hens.

## 2. Results and Discussion

### 2.1. The Effect of Modified Clinoptilolite on Performance of Laying Hens

The effects of modified clinoptilolite (MZ) on the performance of laying hens were separately analyzed, and the results of the analyses are presented in Figure 1, Figure 2 and Figure 3. In this study, OTA decreased the body weight compared to the control and MZ groups (Figure 1). 

At the beginning of the study there was no difference in the average body weight between the treated and control hens. However, during the experiment, particularly from the second week, the body weight of hens in the E-I group tended to be lower (*p* < 0.05) in comparison to all other experimental and control groups. A decrease in body weight associated with a decrease in daily feed consumption was recorded in the E-I group (Figure 2). This result is in agreement with that of Elaroussi et al., 2006 [15], who reported that OTA-associated decreased body weight gain is not caused by the direct effect of OTA, rather is related to reduced feed intake. It is clear that birds from the E-I group might have shown clinical signs of ochratoxicosis. Furthermore, it must also be taken into account that the nutrient absorption capacity of the gastrointestinal tract is probably reduced in these groups. The addition of MZ in the OTA-containing diets (E-III and E-IV) significantly (*p* < 0.05) ameliorated the adverse effects of OTA on body weight. Moreover, body weight in group E-III was 20.72% higher as compared to the control, group E-I. The current findings were similar to the findings of previous researchers, where mineral adsorbent added to diets contaminated with OTA improved the performance of laying hens [14,16,17].

The results presented in Figure 2 show that similar results were observed with the consumption of feed. Compared to all experimental groups and the control group, feed consumption was significantly lower (*p* < 0.05) in the hens fed 1 mg/kg of OTA in their diet (E-I), whereas no significant differences were observed between the control and MZ-treated groups. In this case adding 0.2% of MZ to the feed for laying hens (E-III) had a positive impact on feed consumption (13.5%). This finding is in agreement with the results of several previous studies, which observed that exposing birds to OTA, with minimal amounts of OTA (>0.5 ppm), caused a negative effect on the broiler performance [6]. Although, throughout the study, the hens of the E-II group, which were given the diet with 0.25 mg/kg OTA, had periodically lower feed consumption, no statistically significant differences in the feed consumption were observed among groups that received OTA without (E-II) and with (E-III and E-IV) supplemented MZ. 

The effects of OTA-contaminated diet on average weekly egg production of laying hens over the 49-day period are shown in Figure 3. The present study shows that the daily egg production was significantly reduced (*p* < 0.05) in group E-I, fed with 1 mg/kg of OTA. These negative results persisted throughout the entire experiment. The results of the current experiment suggest that MZ supplementation is required under such circumstances. In this study, OTA at a level of 0.25 mg/kg did not influence egg production. The highest number of eggs was produced by hens which received the commercial diet containing 0.2% MZ. However, there were no statistically significant differences (*p* > 0.05) between the experimental treatments and the control group regarding the number of eggs.

Taken together the current findings imply that the presence of OTA at dietary concentrations of 0.25 μg/kg (E-II) did not influence body weight, feed consumption, or the number of eggs of laying hens. It is necessary to point out that this level is higher than the Serbian regulatory limits for OTA in feed (0.20 mg/kg) for laying hen diets [18] or the EU maximum recommended concentration of 0.1 mg OTA/kg in poultry feed [19]. Results of the current experiment mostly agree with the findings from other studies, where exposure to OTA in low or moderately contaminated feeds did not markedly affect growth performance in poultry [20]. In contrast, the presence of OTA at the concentration of 1 mg/kg in the ingested feed negatively affected laying hen productive performance. It is well known that OTA in ingested feed negatively affects the microvilli of the intestine, including increasing gut permeability, immunity, and bacterial translocation, thereby reducing nutrient absorption [21]. Overall, during the entire experiment, the addition of MZ as a feed additive effectively ameliorated the negative effect of OTA and had significant beneficial effects on body weight, feed consumption, and the number of eggs of laying hens.

The data reported in this study were similar to those obtained by several other authors [2,14,16,22], which confirms that exposure to OTA contaminated feed for a longer period causes a significant decrease in poultry production. Since the gastro-intestinal epithelium is the first barrier meeting mycotoxins, the adverse effect of mycotoxins on the intestinal integrity can be counteracted by the addition of a mycotoxin absorbent. Thus, mycotoxin adsorbents exert an influence on gastro-intestinal health, which can affect nutrient digestibility and therefore improve animal performance and health. The use of clinoptiloite as an absorbent has been extensively studied in Serbia [13,23,24]. Despite other studies that have shown clinoptilolite to be not effective for all mycotoxins or that it might have a negative influence on performance, the present study suggests that clinoptilolite supplementation in poultry feed at the level of 0.2% led to a significant reduction of the harmful effects of OTA, as well as an increase in feed consumption, body weight, and the number of eggs of laying hens.

### 2.2. The Effect of Modified Clinoptilolite on Macroscopic and Pathomorphological Alterations in Target Organs 

The level and duration of exposure are the main predisposing factors regarding the severity of alterations and adverse effects of mycotoxins on poultry health and productivity. The kidney appears to be the primary target organ for OTA toxicity [7,25]. Macroscopic examination showed noticeably enlarged kidneys and livers in hens treated with different amounts of OTA, with a paler color than in other groups. Histopathological examination of kidneys revealed degenerative changes to tubulocytes (cloudy and hydropic degeneration), together with necrobiotic changes (karyopyknosis, karyorrhexis, karyolysis) (Figure 4 and Figure 5). Moreover, tubular atrophy (Figure 6) and cystic dilatation of the tubular lumen and vascular changes (vascular oedema) were observed. All mentioned changes in the kidneys were mainly seen in hens treated only with OTA, while less pronounced but also notable changes were observed in laying hens which had, in addition to OTA in feed, received the adsorbent (Figure 4). Such pathomorphological alterations have also been reported in several studies where the OTA contamination in feed ranged from 0.5–20 mg/kg for a variable time interval [7,26,27].

The incidence and severity of histopathological changes in the liver are presented in Figure 7 and Figure 8. A similar pattern was observed in the liver. Histopathological examination of the liver showed degenerative changes (cloudy swelling, vacuolar degeneration, fatty change), focal necrosis of hepatocytes, and the activation of endothelial and Kupffer’s cells, mainly present in the hens treated with different amounts of OTA. Similarly to in the kidney, the changes were less pronounced in the hens which, in addition to OTA in feed, received the adsorbent, MZ.

Histopathological changes in the kidneys and liver of hens from different experimental groups were in the form of degenerative and necrotic alterations. The examination of kidneys revealed a pattern suggestive of a tubulonephrotic change (cloudy and hydropic degeneration, together with necrobiotic changes; karyopyknosis, karyorrhexis, karyolysis). Beside these changes, tubular atrophy and cystic dilatation of the tubular lumen and vascular changes (vascular edema) were also present. Histopathological examination of the liver tissue also showed degenerative and necrotic changes (cloudy swelling, vacuolar degeneration, fatty change, focal necrosis of hepatocytes, and the activation of endothelial and Kupffer’s cells). All of these pathologic changes observed in the kidneys and liver can be explained by the route of elimination of OTA through the kidneys, and partly through the liver, because of enterophepatic recirculation of this mycotoxin through hepatobiliary excretion. These changes, observed both in the kidneys and liver, were mainly observed in the hens treated only with OTA. On the other hand, the less pronounced changes in the hens that were given the adsorbent can be explained by the protective effect of the adsorbent MZ [19,28,29,30,31,32,33]. It is obvious that the protective effect of the adsorbent MZ depends on the concentration of OTA in the feed. This finding indicates that a dose-dependent study of the protective effects of the adsorbent MZ on different concentrations of OTA in feed should also be performed.

### 2.3. Residue of OTA in Eggs

#### Exposure Assessment and Risk Characterization

Besides the effects on animal health, OTA has the potential to bioaccumulate in the edible tissues of food producing animals, causing food safety issues and posing a hazard to human health. Carry-over studies and surveys of OTA residues in eggs in Serbia have not been extensively carried out. Risk assessment is the systematic characterization of potential adverse effects on humans caused by exposure to hazardous agents. The results of dietary adsorbent MZ on concentrations of OTA residue in eggs and data about the hypothetical daily intake of OTA by consuming contaminated eggs are shown in Table 1. 

It was assumed that both contamination scenarios could occur in reality (E-I and E-II), including the worst-case. Hence, the experiment was performed to determine the protective effects of using adsorbent MZ on different concentrations of OTA residues in eggs. Significant amounts of OTA were detected in the eggs of birds fed with 1 mg/kg OTA (E-I), reaching maximum levels of 0.377 μg/kg (day 49); while supplementing the contaminated diet with adsorbent MZ (1 mg/kg OTA + 0.2% MZ) significantly reduced the residue of OTA in eggs (up to 36.4%). From the above data the carry over rate of OTA was shown to vary between 0.017 and 0.037%. The reason for the maximum level of OTA residue in eggs being on 49 day, presumably lies in the enterohepatic recycling of OTA and consequently its slow elimination [28]. There is a correlation between OTA concentration in feed, duration of intake, and its residues in eggs (r = 0.913, *p* < 0,05). We did not observe OTA residues in the eggs of laying hens fed diets containing 0.25 mg/kg OTA, which confirms the safety of the feed for laying hens from a regulatory point of view. In the avian species, the presence of OTA in eggs has been reported by several authors. Bauer et al. [34] measured OTA concentrations of 0.1–0.2 μg/kg in egg white and of 1.6–4 μg/kg in yolk following the administration of OTA to laying hens at dietary levels of 1.3, 2.6, and 5.2 mg/kg in feed. However, Krogh [35] reported no detection of OTA in eggs of laying hens fed diets containing 0.3 and 1 mg of OTA/kg feed. In contrast, OTA was detected in the eggs of hens fed a higher dosage (10 mg/kg feed) of the toxin [36]. The differences between studies may have been due to the different study settings and methods that were used, and these do not allow for an easy comparison of results. 

Based on the concentrations of OTA determined in egg samples, we roughly estimated the hypothetical dietary intake (EDI) of OTA. For each combination of contamination scenarios, the EDI was calculated. On day 49 of the study, the DI values for the adult population were estimated as 0.195 ng/kgbw/day, for the group that received 1 mg/kg OTA in feed. While, in the second scenario where MZ was added (1 mg/kg OTA + 0.2% MZ) added, the EDI was significantly lower (maximum 0.046 ng/kg w/day). In our study, the calculated EDI and TWI was significantly lower than the tolerable weekly intake proposed by the EFSA (120 ng/kg b.w./week) [37]. Overall, our findings suggest that the public health concern for consuming eggs from these laying hens is negligible. Hence, to ensure the safety of eggs for human consumption, it is extremely important to monitor stored feed in order to provide feed in accordance with the maximum permitted level of OTA in feed for poultry [18,19]. Furthermore, this model can help food safety authorities and feed business operators to better allocate resources for food safety monitoring throughout the food chain.

## 3. Conclusions

The results of a longitudinal study showed that OTA at levels of 1 mg/kg OTA in feed for 7–49 days adversely affected the production performance and resulted in histopathological disturbances to the kidneys and liver of laying hens. Based on these results, the OTA expressed a more or less a negative impact on body weight and feed consumption in both applied concentrations. On the other hand, modified clinoptilolite (MZ) supplementation in poultry feed at the level 0.2% led to a significant reduction in the harmful effects of OTA, on the macroscopic and pathomorphological alterations in target organs, and on production parameters in laying hens. Considering the current risk assessment, the calculated EDI and TWI was significantly lower than the tolerable weekly intake proposed by the EFSA (120 ng/kg b.w./week). Overall, our findings suggest that with this negligible rate of carry-over, no obvious health risks were observed for intake of OTA from egg consumption at the tested concentrations. Despite the low concern, our results indicate a need for continuous monitoring of OTA in feed and a further evaluation of health effects in farm animals in order to avoid or reduce the presence of this natural contaminant in the entire food chain.

## 4. Materials and Methods

### 4.1. Birds and Diets

Forty eight (*n* = 48) laying Lowman Brown hens (27-week-old) were used in this study. The average body weight of birds was 1520 ± 87.29 g, while the study lasted 49 days. At the beginning of the experiment, the number of eggs per housed hen per day was averaged at 0.93 ± 0.3. Birds were placed in a light-controlled (16 h Light:8 h Dark) and temperature-controlled (22 °C) room in wire cages with unlimited access to drinking water.

The OTA-challenge diets were artificially contaminated using OTA produced in vitro by contamination of corn with *Aspergillus ochraceus* Wilhelm NRRL 263.67 culture. Conidospores of *A. ochraceus* were cultivated on potato dextrose-agar substrates for five days at 27 °C, followed by corn contamination with the obtained cultures. The corn with cultures of *A. ochraceus* was kept at a temperature of 15–25 °C during contamination. After 10 days, the corn was dried at 105 °C in a laboratory drying cabinet in order to destroy the mold. The amount of OTA in the ground medium was quantified by LC/MS/MS on the certified standard blank corn. Results showed that the medium contained 270 mg/kg OTA. All other mycotoxins were under the detection limit. After homogenization, the ground medium was mixed into the commercial mixture for laying hens, with 15% of raw proteins. Hens were equally divided into six groups and, depending on the type of addition, were allocated to the following experimental treatments for 7 weeks: E-I group −1 mg/kg OTA; E-II group 0.25 mg/kg OTA; E-III group 1 mg/kg OTA + 0.2% of MZ; E-IV group 0.25 mg/kg OTA + 0.2% of MZ, MZ group of hens was fed with standard diets in addition to only 0.2% of the adsorbent. The control group (C) of hens was fed only a standard diet, without any addition. After mixing the feed, samples were taken in order to determine the level of OTA in the animal feed using the UPLC MS/MS technique. The average concentration of OTA in the feed of the EI group was 1.025 mg/kg, E-II 0.248 mg/kg, E-III 1.011 mg/kg, and E IV −0.251 mg/kg.

Hens were fed once a day, while the standard diet consisted of complete mixtures (with or without the additions above), whose raw and chemical composition was according to the NRC recommendations [38]. This particular adsorbent, which was chosen for its domestic origin, was obtained from the producer for solely scientific purposes.

All procedures were done in accordance with a permit from the Ethics Committee of the Ministry of Agriculture, Forestry, and Water Management, as well as the Veterinary Directorate Republic of Serbia no 323-07-00241/2019-05-01.

### 4.2. Hen Production Performance

Eggs were collected daily to investigate the number of eggs, and the feed consumption was measured daily. On the other hand, body weights were recorded weekly. 

### 4.3. Histopathological Examinations of Livers and Kidneys

The samples for histopathological examination were fixed in a solution of 10% neutral formalin and absolute ethanol and were molded by the standard paraffin technique. Tissue samples (5–8 µm) were then stained by the hematoxylin eosin technique, as described by Scheuder and Chalk [39].

### 4.4. Analysis of Mycotoxins

Before artificial contamination with OTA, the feed was tested for the presence of other mycotoxins (aflatoxin B1, deoxynivalenol, trichothecenes, fumonisins, and zearalenone) in order to avoid synergistic toxic effects on broilers. Analyses of mycotoxins were carried out for OTA as previously published by Nedeljković-Trailović et al. [14].

### 4.5. Residues of OTA in Eggs

Since the OTA levels in eggs were expected to be in the sub-ppb range, a sensitive method for the quantitative determination had to be employed. A QuEChERS-based technique for extraction and clean-up was combined with a highly selective LC-MS/MS analysis to achieve accurate and reliable results.

Analytical standard OTA was purchased from Sigma-Aldrich (St.Louis, MO, USA). Stock solution was prepared in acetonitrile and stored at −20 °C. Acetic and formic acid were obtained from Merck (Darmstadt, Germany). HPLC-grade water, methanol, and acetonitrile were purchased from Sigma-Aldrich. The working standard solution was prepared in acetonitrile by diluting stock solution and stored at 4 °C.

Samples were analyzed with the LC-MS/MS system manufactured by Agilent (Santa Clara, CA, USA), consisting of an Agilent 1290 Infinity II pump, autosampler, and Agilent 6420 Triple quad. The analytical column used for separation was an Agilent Polaris 5, C18 50x2 mm, and 5 µm particle size. Column oven temperature was set at 45 °C. The chromatographic separation was achieved in gradient mode, using water acidified with 0.1% formic acid (mobile phase A) and methanol acidified with 0.1% formic acid (mobile phase B) at a flow rate of 0.35 mL/min. Gradient elution was applied to efficiently separate OTA from matrix interferences. Separation started at 100% A, reaching 100% B in 2 min, and maintaining that composition for another 2 min, followed by three minutes re-equilibration. Total analysis time was 7 min per sample. Electrospray ionization (ESI+) was used, with the following parameters: capillary voltage 4 kV, nebulizer gas (N_2_) pressure was set to 0.35 MPa. Nitrogen was used as a collision gas. The precursor and product ions for OTA were 404 > 239 and 404 > 221, collision energy was 19 V for both transitions, fragment was set to 60 V and cell acceleration voltage to 4 V.

The sample preparation method was modified from the procedure described in the paper presented by Garrido Frenich et al. [40]. Briefly, 5 g of sample, previously homogenized on a Ultra thurrax IKA Yellow line (IKA, Werke, Germany), was weighted into the polypropylene centrifuge tube and 20 mL of methanol/water solution (80/20 *v*/*v*) acidified with 1% acetic acid was added, together with 8 g of sodium acetate anhydrous. This mixture was vortexed for 2 min, followed by shaking in a horizontal IKA Yellow line shaker for 15 min. Tubes were then centrifuged for 5 min at 3500× *g* in a Sigma 2-16P centrifuge (37520 Osterode am Harz, Germany). The cleanup step was performed by passing the previously filtered supernatant through nylon syringe filters (0.45 µm pore size) and through a Waters (Milford, MA, USA) Oasis HLB cartridge. To achieve an increased sensitivity, the purified extract was concentrated under a gentle stream of nitrogen at 50 °C to 0.5 mL and transferred to a HPLC vial for analysis.

The analytical method was validated before the analysis of the samples using blank egg samples, previously determined to be free from OTA, and fortified at various levels. The validation parameters assessed were linearity, limit of quantification, accuracy (expressed as recovery percentage), and precision (expressed as relative standard deviation; RSDr of inter-laboratory reproducibility). Validation levels were from 0.1 to 5 µg/kg, the determined LoQ was 0.05 µg/kg, and the average recovery was from 45% at 0.1 µg/kg to 106% at 5 µg/kg while RSDr at 0.1 µg/kg was 37%. Excellent linearity was observed with an average R^2^ of 0.998.

### 4.6. Calculation of the Human Estimated Daily Intake 

In this study a simplified, hypothetical estimation of human daily intake (EDI) of OTA by consuming eggs contaminated with OTA was used. The daily intake was assessed for different contamination scenarios (different concentrations and durations), combining contamination data from the trial (Σc, μg kg^−1^) with national consumption data (C, kg) and body weight (K) for each consumer according to the following formula [41]: EDI = (Σc) × C/K(1)

The latest assessment of the mean daily intake of eggs is 0.61 pieces/per capita (36.4 g). Mean body weight for the adult Serbian population is considered 70 kg [42]. The individual exposure to OTA was compared with the tolerable weekly intake (TWI) value established by the EFSA (37) (120 ng/kg b.w./week).

### 4.7. Statistical Analysis

All results were statistically analyzed for differences between groups, by an analysis of variance (ANOVA). The results were processed using Graph Pad Prism^®^ 5.0 software (Graph Pad Software Inc., San Diego, CA, USA). All values are expressed as the mean ± SE.

## Figures and Tables

**Figure 1 toxins-13-00469-f001:**
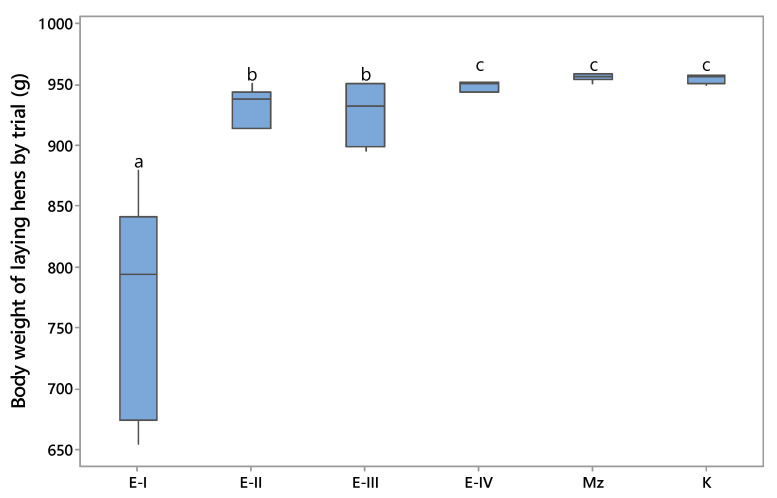
Body weight of laying hens by trial (g) ^a, b, c^—values followed by the same letter superscript are not different at *p* < 0.05 (according to Tukey’s method).

**Figure 2 toxins-13-00469-f002:**
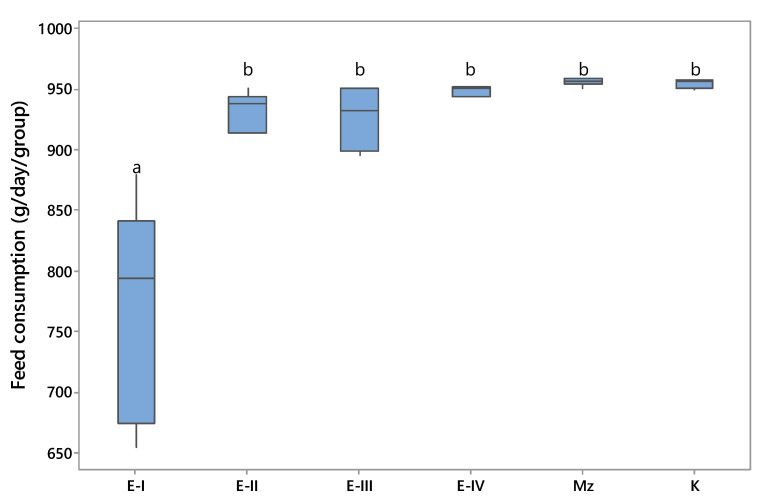
Feed consumption (g/day/per group) ^a, b^—values followed by the same letter superscript are not different at *p* < 0.05 (according to Tukey’s method).

**Figure 3 toxins-13-00469-f003:**
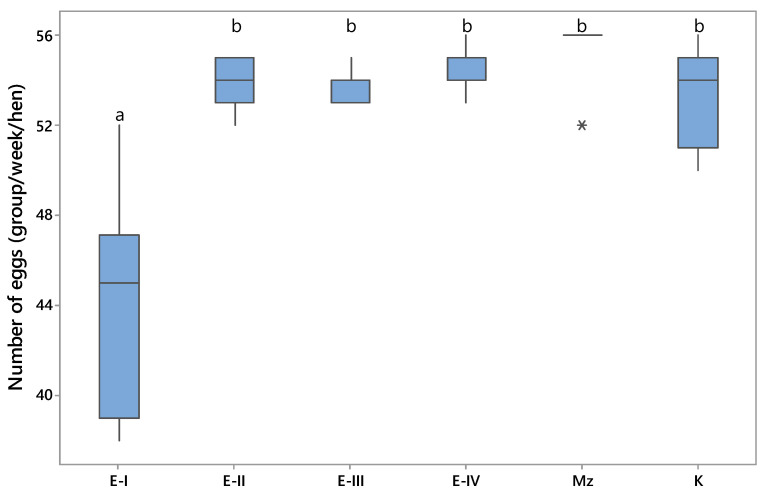
Number of eggs, group/week/hen ^a,b^—values followed by the same letter superscript are not different at *p* < 0.05 (according to Tukey’s method). Asterisk (*) identified the outliers which are observed at least 1.5 times of the interquartile range (Q3–Q1) from the edge of the boxplots.

**Figure 4 toxins-13-00469-f004:**
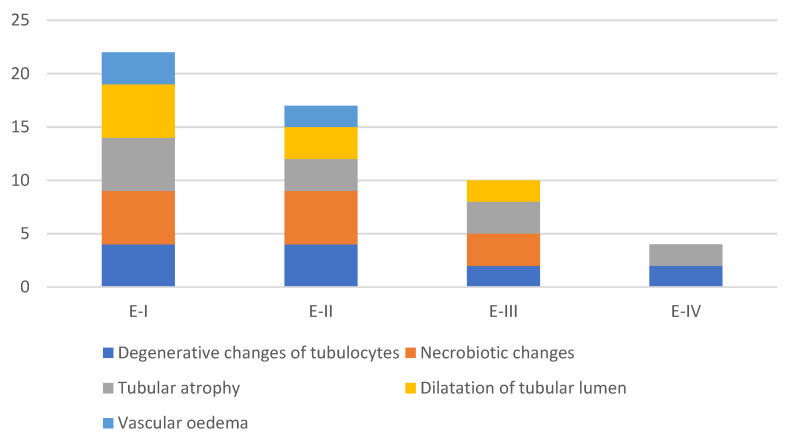
Incidence of histopathological changes on the kidneys.

**Figure 5 toxins-13-00469-f005:**
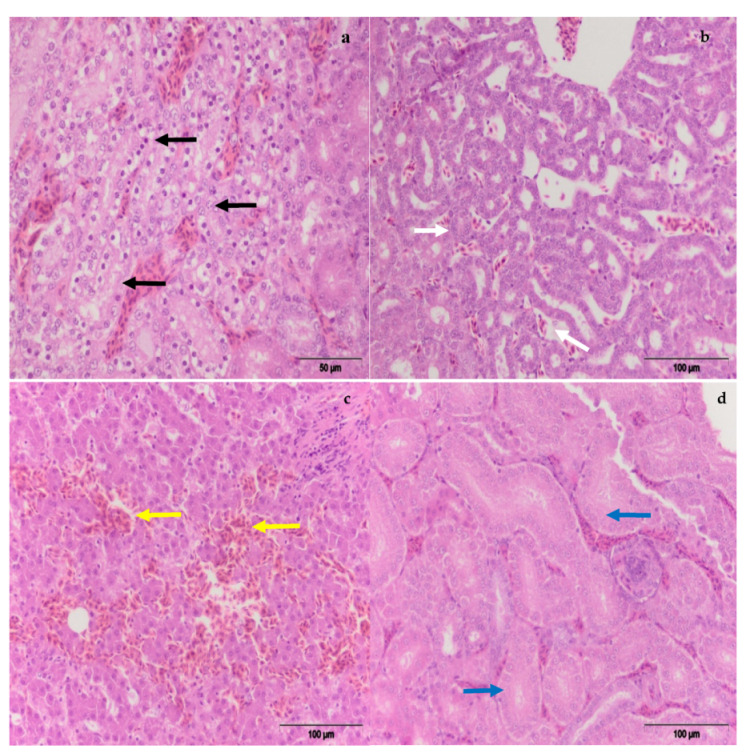
(**a**) Necrobiotic changes (karyopyknosis, karyorrhexis, karyolysis) (black arrows), E-I group; (**b**) tubular atrophy, E-I group (white arrows); (**c**) hemorrhagic areas with massive effusion of red blood cells (yellow arrows), E-I group; (**d**) degenerative changes of tubulocytes (cloudy and hydropic degeneration) E-III group (blue arrows).

**Figure 6 toxins-13-00469-f006:**
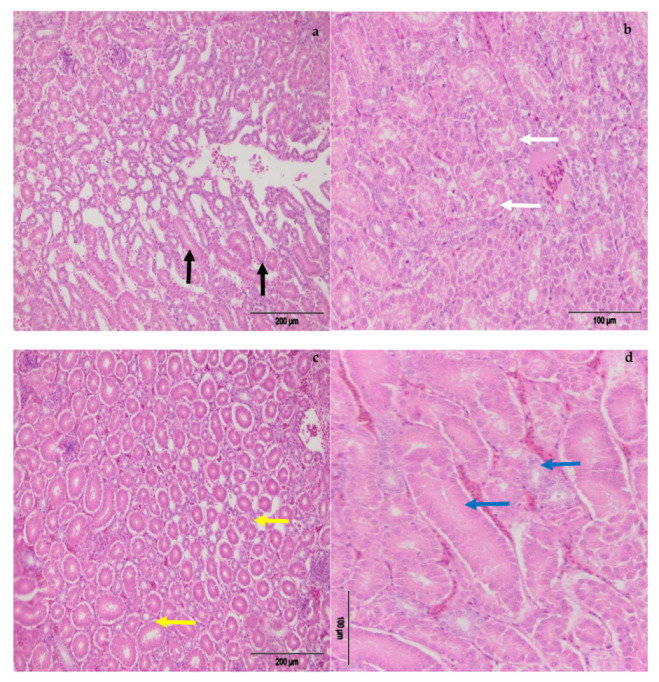
(**a**) Tubular atrophy, E-II group (black arrows); (**b**) degenerative changes of tubulocytes (cloudy and hydropic degeneration) E-II group (white arrows); (**c**) tubular atrophy, E-IV group (yellow arrows); (**d**) degenerative changes of tubulocytes, E-IV group blue arrows).

**Figure 7 toxins-13-00469-f007:**
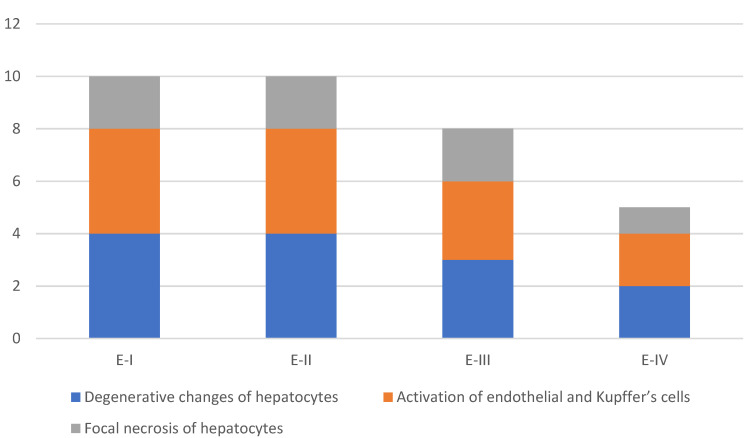
Incidence of histopathological changes in the liver.

**Figure 8 toxins-13-00469-f008:**
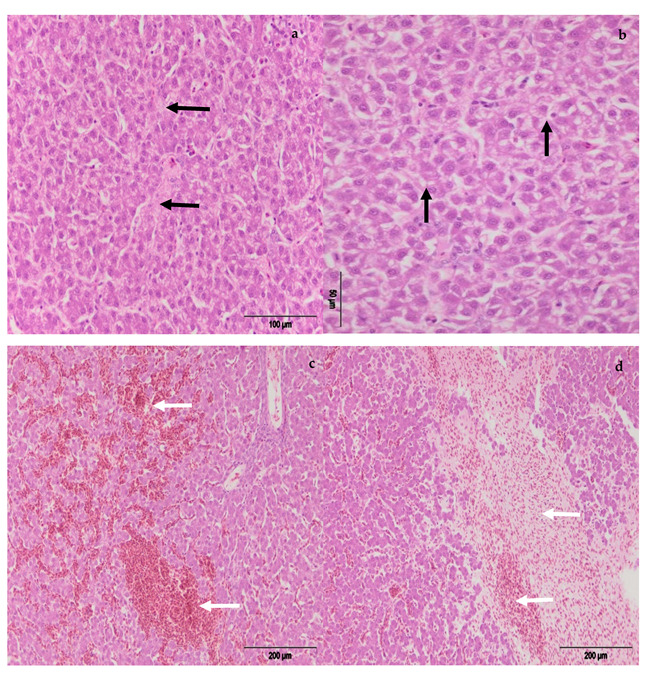
Degenerative changes (vacuolar degeneration, fatty change) of hepatocytes, (**a**) E-I group; (**b**) E-II group (black arrows); liver with hyperemia and hemorrhage, (**c**) E-III group, (**d**) E-IV group (white arrows).

**Table 1 toxins-13-00469-t001:** Effects of dietary adsorbent MZ on concentrations of OTA residue in eggs, and EDI of OTA by consuming contaminated eggs.

Treatment Group					
OTA residue in eggs (μg/kg)	Days	E-I	E-II	E-III	E-IV
	7	0.173 ± 0.00	ND	0.063 ± 0.00	ND
	14	0.157 ± 0.00		0.070 ± 0.00	
	21	0.223 ± 0.02		0.079 ± 0.00	
	42	0.333 ± 0.01		0.084 ± 0.00	
	49	0.377 ± 0.02		0.088 ± 0.00	
EDI (ng/kg bw/day)	7	0.090	ND	0.033	ND
	14	0.082		0.036	
	21	0.116		0.041	
	42	0.173		0.044	
	49	0.196		0.046	

## Data Availability

Not applicable.

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
