# Peer review of "Efficacy of a Modified Clinoptilolite Based Adsorbent in Reducing Detrimental Effects of Ochratoxin A in Laying Hens"

_toxins, 2021, doi:10.3390/toxins13070469_

Round 1

Reviewer 1 Report

Only Fourthly eight laying hens were equally divided into six groups, each group only has 8 birds. The performance data are not reliable. The experimental design has serious flaws. The manuscript should be rejected. 

Author Response

Dear, please find attached email, where are responses to the reviewers comments, according to journal policy.

Reviewer 2 Report

The paper deals with the effect of modified clinoptilolite on the detrimental effects of ochratoxin A in laying hens.

The first part of the paper contains few novelty because the effects of OTA is well known, but the second part, carry over of OTA to egg and calculation of daily human intake is really interesting and important.

The paper is well written, but the figures and photomicrographs requires modification.

Methodically the paper also contains some lackings, such as no data about the measured OTA concentration in the feed of hens in the different groups, and the LOQ and recovery values for the determination of OTA in eggs would be allow only estimation but not correct numerical values.

Some proposals for corrections:

L 17-18: the statement here is valid only for OTA but not for other mycotoxins. Please modify.

L 19: production performance – performance traits is the generally accepted term

L 23: please add the dose of the adsorbent

L 33-35: please add a relevant reference here (e.g., review papers)

L 43: OTA harms not all kind of veterinary therapy but mainly vaccination due to immune-suppression

L 44: 0.5 mg OTA/ kg feed is not minimal because the EU maximum proposed limit for poultry feeds is much lower, 0.1 mg OTA/ kg feed (2006/576/EC)

L 68: such extreme high binding probably true only in pure chemical systems but not in vivo. Please specify the statement

L 80-: Results – this chapter is not only results but also discussion; therefore, should rename the chapter to results and discussion

L 100-102: please modify the graph (Figure 1); columns would be more acceptable than a line

L 103-105: the same problem for Figure 2 as for Figure 1

L 118-120: the same problem for Figure 3 as for Figures 1 & 2

L 133: 0.2 mg OTA/kg feed as the maximum permitted level is specific to Serbia, but the EU proposed a maximum level of 0.1 mg OTA/kg feed as mentioned before

L 174-179: Fig. 5 – please add arrows to the photomicrographs to show the critical degenerative points because most of the readers are not histopathologist

L 207-210: Fig. 8 – the same problem as with Fig. 5

L 231: OTA level mentioned here are not realistic; it should be at mcg/kg but not g/kg level; please double-check the results in the cited reference

L 269: please add the average daily egg production of the laying hens in the study at the start of the experiment

L 281-283: please describe the preparation of the OTA contaminated diets, and add the predicted and measured OTA concentrations in the feeds

L 322: homogenised samples – What it means, whole egg, egg yolk, or egg white?

L 338: the recovery at the range of OTA in the eggs was low (45%), particularly in the group E-III. Therefore the numerical data of the actual OTA concentration in the egg is only estimated value

L 284-285: it would be important to mention here that Minazel Plus was purchased from the producer or obtained for experimental purposes. It is a problem from the point of conflict of interest.

Author Response

Dear, please find attached email, where are responses to the reviewers comments according to journal policy.

Sincerely

Round 2

Reviewer 2 Report

The revised version of the manuscript improved. As the reviewer mentioned in the comments on the original version, the authors modified most of the points. 

My previous opinion about the paper's novelty remains valid for the first part. Still, the second part, about the carryover of OTA to egg and calculation of daily human intake, is new information.

The figures and photomicrographs are accurately modified and just accept.

Methodical part also improved, except in some cases. For instance, there are no data about the predicted and measured OTA content of the diets. The recovery values, in particular in the range of OTA in samples from Group III, remain too low for correct estimation. However, as a theoretical estimation, it can be acceptable.

All of the proposals for editorial changes were modified.

The modified version of the manuscript contains some explanation about obtaining the Minazel Plus product, and I found no conflict of interest.

Author Response

Dear, in enclosed file is response to the reviewer comments. We hope that asnwer is in line with latest reviewer query.

Sincerely Dr Dragan Milicevic
